# Theory of the Kitaev model in a [111] magnetic field

Shang-Shun Zhang [1,2 ✉], Gábor B. Halász[3,4] & Cristian D. Batista [1,5]

Recent numerical studies indicate that the antiferromagnetic Kitaev honeycomb lattice model undergoes a magnetic-field-induced quantum phase transition into a new spin-liquid phase. This intermediate-field phase has been previously characterized as a gapless spin liquid. By implementing a recently developed variational approach based on the exact fractionalized excitations of the zero-field model, we demonstrate that the field-induced spin liquid is gapped and belongs to Kitaev's 16-fold way. Specifically, the low-field non-Abelian liquid with Chern number $C = \pm 1$ transitions into an Abelian liquid with $C = \pm 4$. The critical field and the field-dependent behaviors of key physical quantities are in good quantitative agreement with published numerical results. Furthermore, we derive an effective field theory for the field-induced critical point which readily explains the ostensibly gapless nature of the intermediate-field spin liquid.

[1] Department of Physics and Astronomy, The University of Tennessee, Knoxville, TN 37996, USA. [2] School of Physics and Astronomy and William I. Fine Theoretical Physics Institute, University of Minnesota, Minneapolis, MN 55455, USA. [3] Materials Science and Technology Division, Oak Ridge National Laboratory, Oak Ridge, TN 37831, USA. [4] Quantum Science Center, Oak Ridge, TN 37831, USA. [5] Neutron Scattering Division and Shull-Wollan Center, Oak Ridge National Laboratory, Oak Ridge, TN 37831, USA. ✉email: zhan7927@umn.edu

The exactly solvable Kitaev model on the honeycomb lattice[1] has deepened our insight into quantum spin liquids and helped us in identifying strongly spin-orbit-coupled 4d and 5d materials that may host these exotic quantum phases of matter[2,3]. Indeed, recent years have seen a flurry of such "Kitaev materials" in which the microscopic spin Hamiltonian is believed to approximately realize the Kitaev honeycomb model[4–7]. The most famous ones include the honeycomb iridates, $Na_2IrO_3$[8–13], $\alpha$-$Li_2IrO_3$[14,15], and $H_3LiIr_2O_6$[16], as well as the honeycomb halide $\alpha$-$RuCl_3$[17–25].

While most of these materials are magnetically ordered at the lowest temperatures, the zigzag magnetic order in $\alpha$-$RuCl_3$ can be suppressed with an in-plane magnetic field[26–35]. Also, there are some experimental indications for an intermediate-field spin-liquid phase between the low-field magnetically ordered phase and the high-field spin-polarized phase. Most importantly, a recent experimental work[36] reported a half-integer-quantized thermal Hall conductivity in the intermediate-field regime just beyond the transition out of zigzag order. Though the exact nature of this regime is still an open question, the ongoing experimental efforts reveal the importance of precisely characterizing field-induced spin-liquid phases.

Motivated in large part by the intriguing experimental observations, the behavior of the Kitaev model in a magnetic field has been extensively studied[37] by various approaches, including exact diagonalization[38–42], density-matrix renormalization group (DMRG)[40–43], infinite DMRG (iDMRG)[44], tensor-network methods[45], continuous-time quantum Monte Carlo techniques[46], and slave-particle mean-field theories[47]. These approaches all give consistent results. While the ferromagnetic Kitaev model has a single transition into a polarized phase, the antiferromagnetic Kitaev model includes a new intermediate-field spin liquid between the low-field non-Abelian spin liquid[1] and the high-field polarized phase.

In this work, we implement a novel variational approach[48] to investigate the ground-state phase diagram of the antiferromagnetic Kitaev model in a magnetic field parallel to the [111] direction. This approach is based on the exact fractionalized Majorana-fermion ("spinon") and gauge-flux ("vison") excitations of the pure Kitaev model at zero field[1]. It accounts for two effects of the magnetic field: the renormalization of the Majorana dispersion

through a hybridization with pairs of fluxes (see Fig. 1a) and the finite dispersion acquired by the flux pairs themselves (see Fig. 1b). Remarkably, we find a continuous quantum phase transition, induced by a softening of a hybridized excitation, at a critical field $h_c \simeq 0.50$, which is very close to the critical field $h_c \simeq 0.44$ reported by a recent iDMRG study[44]. The critical point signals the transition of the non-Abelian spin liquid[1] with Chern number $C = \pm 1$ into an Abelian spin liquid with $C = \pm 4$. The predicted field dependence of the flux expectation value and the second derivative of the ground-state energy is also in good quantitative agreement with the iDMRG results. Moreover, the effective field theory of the quantum critical point, as derived from the microscopic Hamiltonian, predicts a low-energy ring of gapped excitations in momentum space, which is difficult to be distinguished from a gapless Fermi surface in finite systems. We conjecture that this is the main reason why previous works[38,40–43] characterized the phase at $h \gtrsim h_c$ as a gapless spin liquid.

## Model

We consider the *antiferromagnetic* Kitaev model[1] in an external magnetic field along the [111] direction,

$$\mathcal{H} = \sum_{\alpha} \sum_{\mathbf{r} \in A} \sigma_{\mathbf{r}}^{\alpha} \sigma_{\mathbf{r}+\hat{\mathbf{r}}_{\alpha}}^{\alpha} + h \sum_{\mathbf{r}} (\sigma_{\mathbf{r}}^x + \sigma_{\mathbf{r}}^y + \sigma_{\mathbf{r}}^z), \quad (1)$$

where $h$ is the magnetic field (in units of the Kitaev energy) and $\hat{\mathbf{r}}_{\alpha}$ is the nearest-neighbor vector from an $A$ site to a $B$ site along an $\alpha$ bond (see Fig. 1). For the exactly solvable Kitaev model in the $h = 0$ limit, the low-energy spectrum comprises gapless matter fermions (i.e., spinons) with a single Dirac cone and gapped dispersionless $\mathbb{Z}_2$ gauge fluxes. These elementary excitations are described in terms of four Majorana fermions $c_{\mathbf{r}}$ and $b_{\mathbf{r}}^{\alpha}$ with $\alpha = x, y, z$ at each site $\mathbf{r}$, where $c_{\mathbf{r}}$ are the matter fermions, and $b_{\mathbf{r}}^{\alpha}$ are bond fermions associated with the $\mathbb{Z}_2$ gauge field $u_{\mathbf{r},\mathbf{r}+\hat{\mathbf{r}}_{\alpha}}^{\alpha} \equiv i b_{\mathbf{r}}^{\alpha} b_{\mathbf{r}+\hat{\mathbf{r}}_{\alpha}}^{\alpha} = \pm 1$. The gauge fields are conserved bond variables that commute with each other; their product around any plaquette $p$ (see Fig. 1a) is gauge invariant and expressible in terms of the physical spins:

$$W_p = u_{12}^z u_{32}^x u_{34}^y u_{54}^z u_{56}^x u_{16}^y = \sigma_1^x \sigma_2^y \sigma_3^z \sigma_4^x \sigma_5^y \sigma_6^z. \quad (2)$$

Thus, $W_p = \pm 1$ can be identified as static $\mathbb{Z}_2$ gauge fluxes. In each flux sector, $\{W_p = \pm 1\}$, represented with an appropriate gauge-field

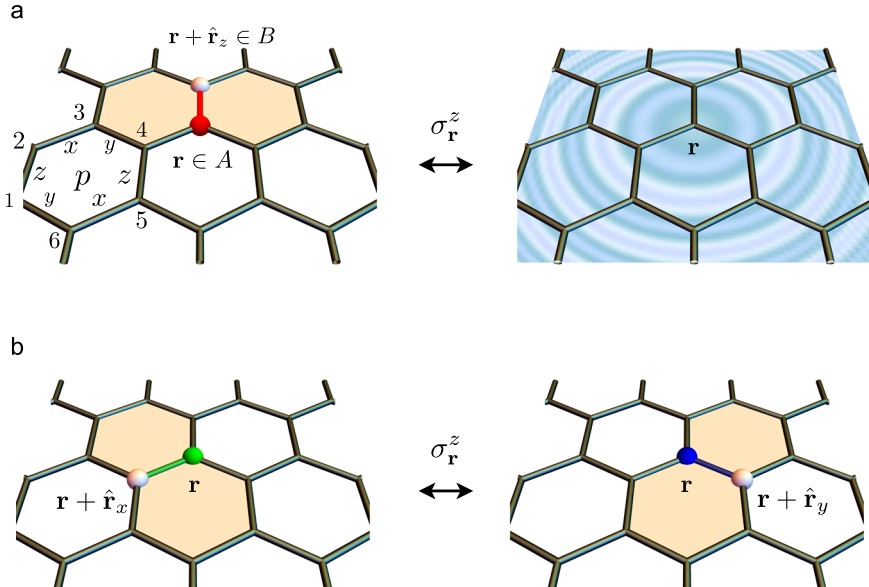

**Fig. 1 Effect of the magnetic field. a** Hybridization between a flux pair and a fermion. **b** Hopping of a flux pair between two neighboring bonds. Definitions of the plaquettes $p$, the lattice sites 1–6, the two sublattices $A$ and $B$, and the nearest-neighbor bond vectors $\hat{\mathbf{r}}_{x,y,z}$ are also shown.

configuration, $\{u^\alpha_{\mathbf{r},\mathbf{r}+\hat{\mathbf{r}}_\alpha} = \pm 1\}$, the zero-field model then reduces to a quadratic matter-fermion problem.

While the model in Eq. (1) is not exactly solvable for a finite field, we can derive a low-energy effective model by projecting $\mathcal{H}$ into the low-energy sector of the pure Kitaev model (corresponding to $h = 0$) generated by single matter-fermion and/or flux-pair excitations[48]. We focus on flux pairs because, unlike single fluxes, they are coherent fermionic quasiparticles[48] and can readily hybridize with matter fermions (see Fig. 1a). The fermionic flux-pair excitations $(\tilde{\chi}^\alpha_{\mathbf{r}\in A})^\dagger = \frac{1}{2}(\tilde{b}^\alpha_{\mathbf{r}} - i\tilde{b}^\alpha_{\mathbf{r}+\hat{\mathbf{r}}_\alpha})$ can be represented with dressed bond-fermion operators $(\tilde{\chi}^\alpha_{\mathbf{r}\in A})^\dagger = \frac{1}{2}(\tilde{b}^\alpha_{\mathbf{r}} - i\tilde{b}^\alpha_{\mathbf{r}+\hat{\mathbf{r}}_\alpha})$ that have the same projective symmetries as the bare bond-fermion operators $(\chi^\alpha_{\mathbf{r}\in A})^\dagger = \frac{1}{2}(b^\alpha_{\mathbf{r}} - ib^\alpha_{\mathbf{r}+\hat{\mathbf{r}}_\alpha})$. The operator $(\tilde{\chi}^\alpha_{\mathbf{r}\in A})^\dagger$ turns the ground state of the pure Kitaev model into an excited state with a single-flux pair on the $\alpha$ bond connected to the site $\mathbf{r} \in A$ by not only creating a bond fermion but also distorting the matter-fermion state: $(\tilde{\chi}^\alpha_{\mathbf{r}\in A})^\dagger |\omega\rangle \otimes |0\rangle = |\phi^\alpha_{\mathbf{r}}\rangle \otimes |\chi^\alpha_{\mathbf{r}}\rangle$, where $|\omega\rangle$ and $|\phi^\alpha_{\mathbf{r}}\rangle$ are the matter-fermion vacua of the gauge-field configurations $|0\rangle$ and $|\chi^\alpha_{\mathbf{r}}\rangle$ that correspond to the flux-free sector and the single-flux-pair sector, respectively. (Mathematically, $|\chi^\alpha_{\mathbf{r}}\rangle = (\chi^\alpha_{\mathbf{r}})^\dagger|0\rangle$, while $|0\rangle$ is the bare-bond-fermion vacuum with $u^\alpha_{\mathbf{r},\mathbf{r}+\hat{\mathbf{r}}_\alpha} = -1$ for all bonds.) If we project the pure Kitaev model [i.e., the first term of Eq. (1)] to its low-energy sector containing at most one matter-fermion or flux-pair excitation, the resulting low-energy Hamiltonian reads

$$\tilde{\mathcal{H}}_{h=0} = \sum_\alpha \sum_{\mathbf{r}\in A} ic_{\mathbf{r}}c_{\mathbf{r}+\hat{\mathbf{r}}_\alpha} + \Delta_\chi \sum_\alpha \sum_{\mathbf{r}\in A} (\tilde{\chi}^\alpha_{\mathbf{r}})^\dagger(\tilde{\chi}^\alpha_{\mathbf{r}}), \quad (3)$$

where the first term is the quadratic matter-fermion problem within the flux-free sector[1], while the second term accounts for the finite energy ($\Delta_\chi \simeq 0.26$) of a flux pair. The Zeeman term [i.e., the second term of Eq. (1)] can then either hybridize a flux pair with a matter fermion (see Fig. 1a) or hop a flux pair to a neighboring bond (see Fig. 1b). By summing $\tilde{\mathcal{H}}_{h=0}$ and the most general symmetry-allowed Hamiltonians describing these two processes, the effective low-energy Hamiltonian for the full model in Eq. (1) becomes

$$\tilde{\mathcal{H}} = \tilde{\mathcal{H}}_{h=0} + h\sum_\alpha \sum_{\mathbf{R}} p_{\mathbf{R},\alpha}\left[\sum_{\mathbf{r}\in A} ib^\alpha_{\mathbf{r}}c_{\mathbf{r}+\mathbf{R}} + \sum_{\mathbf{r}\in B} i\tilde{b}^\alpha_{\mathbf{r}}c_{\mathbf{r}-\mathbf{R}}\right]$$
$$- ihq\sum_{\alpha,\beta} \epsilon_{\alpha\beta}\left\{\sum_{\mathbf{r}\in A} (\tilde{\chi}^\alpha_{\mathbf{r}})^\dagger \tilde{\chi}^\beta_{\mathbf{r}} + \sum_{\mathbf{r}\in B} (\tilde{\chi}^\alpha_{\mathbf{r}-\hat{\mathbf{r}}_\alpha})^\dagger \tilde{\chi}^\beta_{\mathbf{r}-\hat{\mathbf{r}}_\beta}\right\}, \quad (4)$$

where $\mathbf{R}$ is a lattice vector, $\epsilon_{\alpha\beta} = \sum_\gamma \epsilon_{\alpha\beta\gamma}$ is an antisymmetric symbol based on the Levi-Civita symbol $\epsilon_{\alpha\beta\gamma}$, while $p_{\mathbf{R},\alpha}$ and $q$ are dimensionless parameters to be determined. Notice that some $p_{\mathbf{R},\alpha}$ are identical due to the threefold rotation symmetry acting simultaneously in real space and spin space.

Since the effective Hamiltonian $\tilde{\mathcal{H}}$ is quadratic, it can be straightforwardly diagonalized in momentum space:

$$\tilde{\mathcal{H}} = \sum_{\mathbf{k}}\left[i\lambda_{\mathbf{k}} C^\dagger_{\mathbf{k},A} C_{\mathbf{k},B} + \text{H.c.}\right]$$
$$+ \sum_{\mathbf{k},\alpha,\beta}\left\{\Delta_\chi \delta_{\alpha\beta} - ihq\epsilon_{\alpha\beta}\left[1 + e^{i\mathbf{k}\cdot(\hat{\mathbf{r}}_\alpha - \hat{\mathbf{r}}_\beta)}\right]\right\}(\tilde{X}^\alpha_{\mathbf{k}})^\dagger(\tilde{X}^\beta_{\mathbf{k}})$$
$$+ \frac{h}{\sqrt{2}}\sum_{\mathbf{k},\alpha}\left\{iP_{\mathbf{k},\alpha}\left[\tilde{X}^\alpha_{-\mathbf{k}} + (\tilde{X}^\alpha_{\mathbf{k}})^\dagger\right]C_{\mathbf{k},A}\right.$$
$$\left.+ P_{-\mathbf{k},\alpha} e^{i\mathbf{k}\cdot\hat{\mathbf{r}}_\alpha}\left[\tilde{X}^\alpha_{-\mathbf{k}} - (\tilde{X}^\alpha_{\mathbf{k}})^\dagger\right]C_{\mathbf{k},B} + \text{H.c.}\right\}, \quad (5)$$

where $\lambda_{\mathbf{k}} = \sum_\alpha e^{i\mathbf{k}\cdot\hat{\mathbf{r}}_\alpha}$ and $P_{\mathbf{k},\alpha} = \sum_{\mathbf{R}} p_{\mathbf{R},\alpha} e^{i\mathbf{k}\cdot\mathbf{R}}$, while

$$C_{\mathbf{k},\nu} = \frac{1}{\sqrt{2N}}\sum_{\mathbf{r}\in\nu} c_{\mathbf{r}} e^{-i\mathbf{k}\cdot\mathbf{r}}, \quad \tilde{X}^\alpha_{\mathbf{k}} = \frac{1}{\sqrt{N}}\sum_{\mathbf{r}\in A} \tilde{\chi}^\alpha_{\mathbf{r}} e^{-i\mathbf{k}\cdot\mathbf{r}} \quad (6)$$

are momentum-space matter and bond fermions in terms of the sublattice index $\nu = A, B$ and the system size $N$. By considering

the matrix elements of the Zeeman term $\propto h$ in Eq. (1) within the low-energy sector of the pure Kitaev model[48], we relate the dimensionless parameters in Eq. (5) to matter-fermion matrix elements of this exactly solvable model (Note: see the Supplementary Information for more details on the dimensionless parameters of the effective Hamiltonian, the expectation value of the flux operator, the coefficients of the effective field theory, and the nonanalytic behavior of the ground-state energy):

$$q = \langle\phi^\beta_{\mathbf{0}}|(1 + ic_0 c_{\hat{\mathbf{r}}_\alpha})|\phi^\gamma_{\mathbf{0}}\rangle, \quad \alpha \neq \beta \neq \gamma,$$
$$P_{\mathbf{k},\alpha} = \langle\phi^\alpha_{\mathbf{0}}|\omega\rangle + \frac{1}{2}\sum_{\mathbf{k}'}(1 - e^{-i\mathbf{k}'\cdot\hat{\mathbf{r}}_\alpha + i\varphi_{\mathbf{k}'}})\langle\phi^\alpha_{\mathbf{0}}|\psi^\dagger_{\mathbf{k}'}\psi^\dagger_{\mathbf{k}}|\omega\rangle, \quad (7)$$

where $\mathbf{r} = \mathbf{0}$ is an $A$ site, while $\psi_{\mathbf{k}} = (C_{\mathbf{k},A} + ie^{i\varphi_{\mathbf{k}}}C_{\mathbf{k},B})/\sqrt{2}$ in terms of $e^{i\varphi_{\mathbf{k}}} = \lambda_{\mathbf{k}}/|\lambda_{\mathbf{k}}|$ are the matter fermions diagonalizing the flux-free sector of the pure Kitaev model. For a finite honeycomb lattice with $N = 121 \times 121$ unit cells, we numerically find $q \simeq 0.0494$ and $P_{\mathbf{0},\alpha} \simeq 0.722$.

## Results

We study the low-energy effective model in Eq. (4) as a function of the magnetic field $h$. At zero field, the spectrum coincides with that of the pure Kitaev model and contains one dispersive matter-fermion band as well as the three flat bond-fermion bands (see Fig. 2a). For a small field, $h \ll \Delta_\chi$, the hybridization between these four bands gives rise to a finite energy gap, $\Delta_K(h) \propto h^3$, at the K point of the Brillouin zone (BZ). The slow field dependence of $\Delta_K(h)$, which is expected from a perturbative argument by Kitaev[1], explains why the global minimum of the band structure remains at the K point up to a large field, $h_0 \simeq 0.46$. As shown in Fig. 3a, the global minimum switches from the K point to the $\Gamma$ point at $h = h_0$, and the corresponding gap, $\Delta_\Gamma(h)$, closes at a slightly larger field, $h_c \simeq 0.50$ (see Fig. 2b). Since the little group of the $\Gamma$ point includes the threefold rotation $C_3$, the fermion eigenmodes at the $\Gamma$ point can be classified according to their $C_3$ eigenvalues. The natural bond-fermion modes, corresponding to $C_3$ eigenvalues 1 and $e^{\mp 2\pi i/3}$, respectively, are then

$$\tilde{X}^0_{\mathbf{0}} = \left(\tilde{X}^x_{\mathbf{0}} + \tilde{X}^y_{\mathbf{0}} + \tilde{X}^z_{\mathbf{0}}\right)/\sqrt{3},$$
$$\tilde{X}^\pm_{\mathbf{0}} = \left(\tilde{X}^x_{\mathbf{0}} + e^{\pm 2\pi i/3}\tilde{X}^y_{\mathbf{0}} + e^{\mp 2\pi i/3}\tilde{X}^z_{\mathbf{0}}\right)/\sqrt{3}. \quad (8)$$

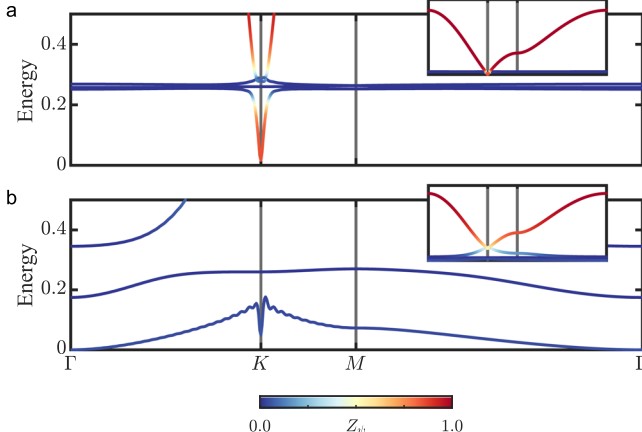

**Fig. 2 Low-energy spectrum of the effective Hamiltonian.** The fermion dispersions correspond to $h = 0.05$ in (**a**) and $h = h_c \simeq 0.50$ in (**b**). The color scale shows the matter-fermion weight, $0 < Z_\psi < 1$, of the given fermion eigenmode; red (blue) color indicates predominantly matter-fermion (bond-fermion) character. The insets show the spectrum over the full energy range. Note that a hybridization decay length, $\xi = 25$, is used to regularize the K-point behavior (See the "Note" above earlier).

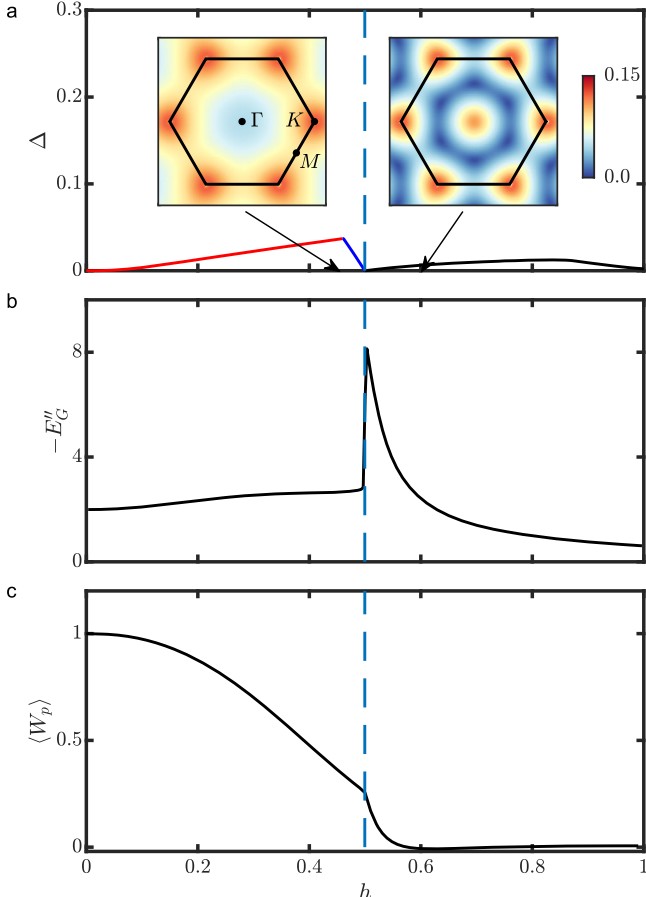

**Fig. 3 Field dependence of key physical quantities. a** Overall energy gap. The insets show the dispersion of the low-energy fermion eigenmode on both sides of the phase transition, with the black solid hexagon marking the Brillouin zone. The red (blue) line corresponds to the gap $\Delta_K$ ($\Delta_\Gamma$), while the black line corresponds to the gap at the six wave vectors $\boldsymbol{Q}_j$, the corners of the blue hexagon in the right-hand-side inset. **b** Second derivative of the ground-state energy. **c** Expectation value of the $\mathbb{Z}_2$ gauge flux.

Since the matter-fermion mode $\psi_{\boldsymbol 0}$ is invariant under $C_3$, it can only hybridize with the bond-fermion mode $\tilde X_{\boldsymbol 0}^0$. At the critical field, $h_c = 3\sqrt{\Delta_\chi/2}\,(\sum_\alpha P_{\boldsymbol 0,\alpha})^{-1} \simeq 0.50$, one of the resulting hybridized eigenmodes is gapless. In contrast, there is a higher critical field, $h'_c = \Delta_\chi/(2\sqrt 3 q) \simeq 1.52$ (not shown in Fig. 3), at which the pure bond-fermion eigenmode $\tilde X_{\boldsymbol 0}^+$ has vanishing energy. We note that a complete diagonalization over the full BZ reveals yet another critical point at $h''_c \simeq 1.0$ due to the softening of a hybridized mode at the M point. We emphasize, however, that the effective model is no longer expected to be valid when $h$ is significantly larger than $h_c$.

Figure 3a shows the overall energy gap as a function of the magnetic field $h$. As expected, the gap is proportional to $h^3$ at the smallest fields, $h \ll \Delta_\chi$. Just below $h_c$, the global minimum of the excitation spectrum switches from the K point to the Γ point, and the gap vanishes at $h_c \simeq 0.50$[38,40,44]. Importantly, the zero-energy mode at $h=h_c$ has dominant bond-fermion character with a large bond-fermion weight $6/(6+\Delta_\chi) \simeq 0.96$ (see also Fig. 2b), which is consistent with the numerical closing of the vison gap in the specific heat[38]. In contrast, the gap reopens for $h \gtrsim h_c$, which appears to be in contradiction with the same numerical results and the corresponding conjecture of a gapless U(1) spin liquid at intermediate fields. However, our analytic approach can also

explain the numerical similarity between the gapped spin liquid at $h \gtrsim h_c$ and a gapless spin liquid with a circular spinon Fermi surface. Indeed, as we explain below, the phase transition at $h = h_c$ gives rise to a low-energy ring at $h \gtrsim h_c$ (see the inset of Fig. 3a) which expands from the Γ point and corresponds to a small energy gap $\propto (h - h_c)^{3/2}$. This low-energy ring naturally explains the large low-energy density of states found by exact diagonalization[38,40]. The emergence of the low-energy ring and the nature of the $h \gtrsim h_c$ phase are explained in the next section, where we derive an effective field theory to describe the continuous topological phase transition at $h = h_c$.

Figures 3b and c plot the second derivative of the ground-state energy, $E''_G = \mathrm{d}^2 E_G/\mathrm{d}h^2$, and the expectation value of the $\mathbb{Z}_2$ gauge flux, $\langle W_p \rangle$, against the magnetic field. As we explain below, the discontinuity of $E''_G$ at $h = h_c$ is a generic property of the corresponding phase transition. This discontinuity leads to a peak in $E''_G$ at $h = h_c$, which is qualitatively and quantitatively consistent with the iDMRG results[44]. We note that our result for $\langle W_p \rangle$ (See the "Note" above earlier) (see Fig. 3c) is also consistent with iDMRG.

We argue that our effective model in Eq. (4) remains valid up to a field $h \gtrsim h_c$ just beyond the first phase transition. Indeed, the fractionalized excitations of the pure Kitaev model remain well defined throughout the low-field phase at $h < h_c$; however, after the first phase transition induced by their softening, these original excitations are superseded by the emergent excitations of the higher-field phase. Therefore, we focus on the first phase transition at $h = h_c$ throughout the rest of this work.

Remarkably, the critical field $h_c \simeq 0.50$ is only 10% higher than the corresponding iDMRG result, $h_c \simeq 0.44$[44]. Also, the slight overestimation of $h_c$ is not surprising because the inclusion of higher-energy ($E \simeq 2\Delta_\chi$) states with four fluxes and one matter fermion would lead to a reduction of $h_c$. Finally, at $h = h_c$, the dynamical spin structure factor from iDMRG indicates that the spin excitation gap closes at the Γ point, which is in agreement with our results. Indeed, since a spin excitation fractionalizes into a pair of fermion excitations, and the fermions at $h = h_c$ are gapless at the Γ point (see Fig. 2b), a pair of gapless fermions has zero total momentum, corresponding to a vanishing spin gap at the Γ point. These similarities between the iDMRG results and those obtained from our effective Hamiltonian $\tilde{\mathcal H}$ indicate that our variational low-energy manifold captures the essence of the phase transition at $h = h_c$ and the new spin-liquid phase at $h \gtrsim h_c$.

**Field theory of topological phase transition.** In the vicinity of the critical field, $h \simeq h_c \simeq 0.50$, the low-energy fermion eigenmodes belong to the trivial representation of $C_3$, and the long-wavelength limit of $\tilde{\mathcal H}$, corresponding to the region around the Γ point, can be written as

$$\tilde{\mathcal H}_{\mathrm{eff}} = \sum_{\boldsymbol k} f_{\boldsymbol k}^\dagger [\beta_{\boldsymbol k}^x \tau_x + \beta_{\boldsymbol k}^y \tau_y + \beta_{\boldsymbol k}^z \tau_z] f_{\boldsymbol k}, \qquad (9)$$

where $\tau_{x,y,z}$ are the Pauli matrices, and $f_{\boldsymbol k} = (f_{1,\boldsymbol k}, f_{2,\boldsymbol k})^{\mathrm T}$ is a two-component fermionic operator corresponding to the two zero-energy modes of $\tilde{\mathcal H}$ at the critical field:

$$f_{1,\boldsymbol k} = \sqrt{\frac{6}{6+\Delta_\chi}}\,\tilde X_{\boldsymbol k}^0 - i\sqrt{\frac{\Delta_\chi}{6+\Delta_\chi}}\,\psi_{\boldsymbol k}, \quad f_{2,\boldsymbol k} = f_{1,-\boldsymbol k}^\dagger. \qquad (10)$$

The coefficients $\beta_{\boldsymbol k}^{x,y,z}$ in Eq. (9) must be $C_3$ invariant real polynomials. Up to cubic order in $\boldsymbol k$, there are only four such polynomials: the trivial polynomial 1, the quadratic polynomial $k^2 = k_x^2 + k_y^2$, and the cubic polynomials $g_{\boldsymbol k}^x = k_x(3k_y^2 - k_x^2)$ and $g_{\boldsymbol k}^y = k_y(3k_x^2 - k_y^2)$. Moreover, the particle-hole symmetry of the original Hamiltonian $\mathcal H$ dictates that $\tilde{\mathcal H}_{\mathrm{eff}}$ must remain invariant

under $f_{\mathbf{k}} \to \tau_x (f^{\dagger}_{-\mathbf{k}})^{\mathrm{T}}$, implying that the polynomials $\beta^{\mu}_{\mathbf{k}}$ must satisfy the following relationships:

$$\beta^x_{\mathbf{k}} = -\beta^x_{-\mathbf{k}}, \ \beta^y_{\mathbf{k}} = -\beta^y_{-\mathbf{k}}, \ \beta^z_{\mathbf{k}} = \beta^z_{-\mathbf{k}}. \quad (11)$$

These symmetry considerations then lead to the general forms

$$\beta^z_{\mathbf{k}} = c_0 + c_z k^2,$$
$$\beta^{\eta}_{\mathbf{k}} = \sum_{\nu = x,y} c_{\eta\nu} g^{\nu}_{\mathbf{k}}, \quad \eta = x, y, \quad (12)$$

where $c_0$, $c_z$, and $c_{\eta\nu}$ are, in general, functions of $h$. Since the phase transition at $h = h_c$ is driven by a sign change in $c_0$, we assume that $c_z$ and $c_{\eta\nu}$ are constants, while we write $c_0 = c'_0 (h - h_c)$ with a constant $c'_0$. Starting from Eqs. (5) and (7), and defining all lengths in units of the lattice vector (i.e., the distance between two neighboring $A$ sites), the constants are derived to be $c'_0 \simeq -1.00$, $c_z \simeq 0.0125$, $c_{xx} \simeq -0.00268$, $c_{yy} \simeq -0.00088$, and $c_{xy} = c_{yx} = 0$ (See the "Note" above earlier). Then, using Eq. (9), the fermion dispersion is given by

$$\omega_{\mathbf{k}} = \sqrt{(\beta^x_{\mathbf{k}})^2 + (\beta^y_{\mathbf{k}})^2 + (\beta^z_{\mathbf{k}})^2} \quad (13)$$

and becomes gapless at $\mathbf{k} = \mathbf{0}$ for $h = h_c$. For $h < h_c$, the dispersion is dominated by the function $\beta^z_{\mathbf{k}}$ and is largely quadratic: $\omega_{\mathbf{k}} \simeq |c'_0|(h_c - h) + c_z k^2$. In contrast, for $h > h_c$, the function $\beta^z_{\mathbf{k}}$ vanishes for $|\mathbf{k}| = \sqrt{|c'_0|(h - h_c)/c_z}$. Thus, along this ring of radius $|\mathbf{k}|$, the energy gap is determined by the small cubic contributions from $\beta^{x,y}_{\mathbf{k}}$ and has a slow field dependence: $\Delta \propto (h - h_c)^{3/2}$. The net result is a ring of low-energy fermions around the $\Gamma$ point (see the inset of Fig. 3a).

The effective field theory in Eq. (9) describes a continuous topological phase transition. The phases on both sides of the transition belong to Kitaev's 16-fold way[1] and are characterized by the fermion Chern number. The contribution from the low-energy fermions to this Chern number is given by[49]

$$C = \frac{1}{4\pi} \int d\mathbf{k} \ \mathbf{d}_{\mathbf{k}} \cdot [\partial_{k_x} \mathbf{d}_{\mathbf{k}} \times \partial_{k_y} \mathbf{d}_{\mathbf{k}}], \quad (14)$$

where $\mathbf{d}_{\mathbf{k}} = \boldsymbol{\beta}_{\mathbf{k}}/|\boldsymbol{\beta}_{\mathbf{k}}|$ and $\boldsymbol{\beta}_{\mathbf{k}} = (\beta^x_{\mathbf{k}}, \beta^y_{\mathbf{k}}, \beta^z_{\mathbf{k}})$. Geometrically, $C$ is simply the skyrmion number of the vector field $\mathbf{d}_{\mathbf{k}}$. Figure 4 depicts the vector field $\mathbf{d}_{\mathbf{k}}$ around the $\Gamma$ point on both sides of the phase transition at $h = h_c$. While the field configuration is topologically trivial for $h < h_c$, it includes six merons (three skyrmions) for $h > h_c$. The corresponding change in the Chern number, $\Delta C = 3$, is then a generic property of the phase transition described by $\tilde{\mathcal{H}}_{\mathrm{eff}}$. To understand the emergence of the six merons around the $\Gamma$ point, we first note that $\beta^{\eta}_{\mathbf{k}} \propto \mathrm{Im}(k^3_+ e^{-i\phi_{\eta}})$ with $k_+ = k_x + i k_y$ and $\phi_{\eta} = \arctan(c_{\eta x}/c_{\eta y})$. Each function $\beta^{\eta}_{\mathbf{k}}$ (with $\eta = x, y$) possesses three nodal lines corresponding to

$k_y/k_x = \tan(\phi_{\eta}/3 + \varphi)$ with $\varphi = 0, \pi/3, 2\pi/3$. Ignoring the $\beta^y_{\mathbf{k}}$ function, the low-energy spectrum then contains six Dirac nodes $\mathbf{Q}_j$ (with $j = 1, 2, \ldots, 6$) at the intersections of the nodal lines of $\beta^x_{\mathbf{k}}$ and the ring of radius $|\mathbf{k}| = \sqrt{|c'_0|(h - h_c)/c_z}$. The vorticity of the vector field $\mathbf{d}_{\mathbf{k}}$ around each Dirac node $\mathbf{Q}_j$ is $(-1)^j$. Assuming $\phi_x \neq \phi_y$ (which is true in our case), the finite value of $\beta^y_{\mathbf{Q}_j} \propto (-1)^j$ generates a mass term for each Dirac node in such a way that the Dirac nodes all give identical contributions ($+1/2$ each or $-1/2$ each) to the change in the Chern number. The net change in the Chern number is then

$$\Delta C = 3 \ \mathrm{sgn}\left[\det \hat{\mathcal{C}}\right], \quad \hat{\mathcal{C}} = \begin{pmatrix} c_{xx} & c_{xy} & 0 \\ c_{yx} & c_{yy} & 0 \\ 0 & 0 & c_z \end{pmatrix}. \quad (15)$$

Using the constants $c_z$ and $c_{\eta\nu}$ given above, we obtain $\Delta C = 3$ at the critical field $h = h_c$. Since the low-field phase at $h < h_c$ is well known[1] to have Chern number 1, we conclude that the higher-field phase at $h \gtrsim h_c$ has Chern number 4.

We next consider the second derivative of the ground-state energy $E''_G$ with respect to the magnetic field $h$. The universal critical behavior at $h = h_c$ is determined by the low-energy modes $|\mathbf{k}| \leq \Lambda$, where the cutoff $\Lambda$ can be made arbitrarily small (corresponding to an infrared singularity). While the contribution of these modes to $E''_G$ is $\propto \Lambda^2$ for $h \to h^-_c$, it is an $\mathcal{O}(1)$ constant for $h \to h^+_c$. In particular, there is a contribution from the neighborhood of the low-energy ring at $h \gtrsim h_c$ which is independent of the cutoff $\Lambda$. Therefore, we obtain a discontinuity in $E''_G$ at the critical field (See the "Note" above earlier):

$$\Delta E''_G = \lim_{h \to h^-_c} E''_G - \lim_{h \to h^+_c} E''_G = \frac{\sqrt{3}(c'_0)^2}{8\pi c_z}. \quad (16)$$

Remarkably, this discontinuity in $E''_G$, as shown in Fig. 3b, is entirely determined by two coefficients of the effective field theory. From the constants $c'_0$ and $c_z$ given above, it is found to be $\Delta E''_G \simeq 5.5$, which is consistent with the corresponding result for a finite lattice (see Fig. 3b). The quantitative agreement between this value and the one obtained from iDMRG[44] indicates that the effective field theory at $h = h_c$ is both qualitatively correct and quantitatively accurate.

## Discussion

Our simple and accurate variational approach to extended Kitaev models[48] indicates that the antiferromagnetic (AFM) Kitaev model undergoes a continuous quantum phase transition driven by a magnetic field parallel to the [111] direction. According to this approach, the new phase, which has been reported in previous numerical works[38–44], is a gapped chiral spin liquid with a ring of low-energy excitations. Due to its large low-energy density of states, it is difficult for numerical simulations to distinguish this low-energy ring from a gapless Fermi surface. In particular, while DMRG may, in principle, detect gapless modes via a finite value of the central charge[41–44], different studies find conflicting values[43] or even unphysical non-integer values[44], thereby indicating that the currently available system sizes cannot be used to determine whether the new phase is gapped or gapless[44].

In contrast to the non-Abelian low-field phase, the new phase at higher fields possesses Abelian topological order with four distinct types of anyons: 1 (vacuum), $\varepsilon$ (fermion), as well as $e$ and $m$ (vortices). The two phases can then be distinguished numerically by computing the entanglement spectrum[50] or the topological entanglement entropy for a bipartition of an infinite cylinder[51–53], readily available in iDMRG[44]. However, due to the

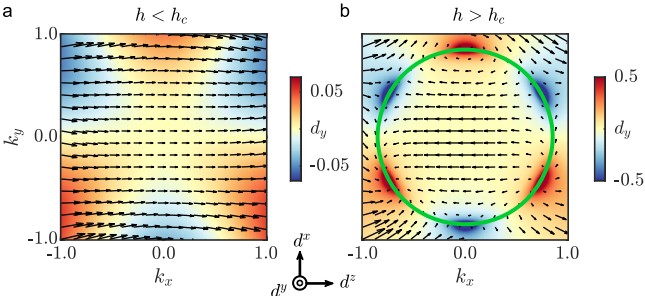

**Fig. 4 Topological phase transition.** Configuration of the unit-vector field $\mathbf{d}_{\mathbf{k}}$ on the two sides of the phase transition, **a** $h < h_c$ and **b** $h > h_c$. The color scale shows the component $d^y_{\mathbf{k}}$, while the black arrows represent the components $(d^z_{\mathbf{k}}, d^x_{\mathbf{k}})$. The green circle marks the low-energy ring.

challenges mentioned above, such a numerical confirmation of our predictions may require the addition of irrelevant Hamiltonian terms that increase the gap in the higher-field phase without generating new phase transitions.

From an experimental perspective, it is important to note that the higher-field spin liquid is known to be stable against both Heisenberg and Gamma interactions[38], making it more likely to emerge in real materials. Also, in the presence of ferromagnetic Heisenberg terms, a field-induced transition between the higher-field spin liquid and a lower-field zigzag order, potentially relevant for $\alpha$-RuCl$_3$, has been reported[42]. According to our theory, the key experimental signature of the higher-field spin liquid is a specific quantized value of the thermal Hall conductivity, $\kappa_{xy} = \pi k_B^2 T/(3\hbar)$, which is four times larger than for the low-field non-Abelian spin liquid.

We next remark that our variational approach is still approximately valid in the presence of both a matter-fermion *and* a flux-pair excitation and that, in the presence of non-Kitaev interactions, it can also be used to describe *bound states* between these two types of excitations[48]. Since such a bound state corresponds to a spin excitation, its softening leads to a divergent magnetic susceptibility for some wave vector and thus signals the onset of magnetic ordering.

We also emphasize that our approach straightforwardly generalizes to the ferromagnetic (FM) Kitaev model. In this case, the first term in Eq. (3) has a negative sign, and the flux-pair-hopping parameter in Eq. (7) is found to be $q \simeq 1.35$, i.e., about 30 times larger than for the AFM Kitaev model. Therefore, the lowest-field phase transition is driven by a softening of a pure flux-pair mode and happens at a much smaller critical field, $h' = \Delta_\chi/(2\sqrt{3}q) \simeq 0.056$. The strong asymmetry between the FM and AFM Kitaev models is due to opposite (constructive and destructive) interference effects between the two processes contributing to flux-pair hopping[48]. We note that this asymmetry is not apparent in the simplified perturbative analysis of ref. [1] because it neglects the energy dispersions of the intermediate states. We further remark that our results for the FM Kitaev model are also consistent with numerical studies that report a single first-order transition into a trivial polarized phase at a critical field $h_p \simeq 0.028$[44]. At this first-order phase transition, corresponding to $h_p \lesssim h'$, the fluxes suddenly proliferate and confine all fractionalized excitations.

Finally, going back to the AFM Kitaev model, it is interesting to note that a recent work[54] has also found a field-induced chiral spin liquid phase with Chern number $C = 4$ through a completely different approach.

## Data availability
The data that support the findings of this study are available from the corresponding authors upon reasonable request.

## Code availability
The codes that support the findings of this study are available from the corresponding authors upon reasonable request.

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

## Acknowledgements
The authors thank Matthias Gohlke, Frank Pollmann, and Federico Becca for useful discussions. S-S.Z. and C.D.B. are supported by funding from the Lincoln Chair of Excellence in Physics. G.B.H. was supported by the U.S. Department of Energy, Office of Science, National Quantum Information Science Research Centers, Quantum Science Center.

## Author contributions
All authors, S-S.Z, G.B.H., and C.D.B., made significant contributions to the manuscript.

## Competing interests
The authors declare no competing interests.
