## [Peer Review File · Nature Communications]

REVIEWER COMMENTS

Reviewer #1 (Remarks to the Author):

Authors studied the phase transition of antiferromagnetic Kitaev model under the magnetic field perpendicular to the honeycomb plane, [111]-axis. This problem (and extended model) was studied earlier by several others (Refs.37 - 45) using numerical methods or slave particle theories. Authors found an intermediate phase under the field, similar to other previous studies, but the nature of the intermediate phase is different from the earlier claims. They report that the intermediate phase is a gapped spin liquid, which belongs to Kitaev's 16-fold way. The Chern number changes from ± 1 to ± 4 , across the transition around $h_c \sim 0.5$, and it is a continuous transition.

I find the result interesting if true, but unfortunately it is very difficult to read the draft and verify the conclusion. Furthermore, the physical origin of the main conclusion is not clear as the draft is not self-contained. For example, Eq. (3) is a low-energy effective Hamiltonian, but the explanation on how such Hamiltonian is derived in the manuscript is minimal (maybe due to the page limit of the journal). Since authors refer to Ref. 46 (arXiv:2013.13274), which is another paper by the authors, I had to read the reference to get a rough idea on the Hamiltonian. Ref. 46 shows how other interactions such as Heisenberg and Gamma terms lead to various magnetic ordering states.

Few questions need to be addressed, before I make any recommendation. 1. How can one be sure that the gapped spin liquid phase is not preempted by another phase? For example, numerical studies found a putative gapless spin liquid around $h_c \sim 0.44$. It is possible that this gapless phase is different from the author's gapped phase, and preempt the claimed gapped phase. Is there a reason why such scenario is impossible?

2. The current theory cannot capture the magnetically polarized phase, the most trivial phase when the field is very large. If so, how one could confirm that the phase does exist? The theory (Eq. 3) may breakdown before h reaches h_c . Could authors quantify a critical strength beyond which the theory does not apply and the gapped phase is below the critical field?

3. Why the magnetic field acts different from other interactions? Eventually the magnetic field leads to the polarized phase, so if such a possibility (polarized phase) is included together with an incommensurate ordering, can one get an incommensurate (IC) ordering in addition to the gapped spin liquid, or the IC preempts the spin liquid?

In summary, I found the result interesting, but I cannot recommend the current version of the draft for publication.

Reviewer #2 (Remarks to the Author):

The paper aims to provide an interpretation and explanation of previous numerical results concerning the Kitaev model in a magnetic field. Motivated by experimental reports that a magnetic field can suppress magnetic ordering in α - RuCl_3 and that a topological spin liquid may then replace it, several groups had studied the Kitaev model (and its variants) using numerical approaches. These had found an intermediate phase between the Kitaev spin liquid (at zero field) and the magnetically polarized regime.

The present paper presents an analytic (but approximate) treatment of this transition in order to provide insights beyond the numerical findings. Results compare well with numerics and the biggest difference -- gapped excitations, where numerics have not found a gap -- are reasonably explained by the gaps

small size.

The point of this work is that this analytic treatment yields insights that are not easily obtained using numerics (or experiments), e.g. a Chern number of 4 and the fact that it is Abelian. The work appears to be carefully carried out and correct, it is also reasonably clearly described. Given that the study of (extended) Kitaev models continues to be a vibrant field of research, where analytic results are however much rarer than numerical ones, these findings will be of strong interest to readers.

It is not so clear to me to what extent the paper will inspire new work beyond the immediate community working on Kitaev models. One thing that could improve the manuscript would be if the conclusions could pick up a theme strongly present in the introduction, namely the experimental research. It would be nice to read a discussion not only of the more immediate

relation to the numerical work, but also to experiment. (E.g. the field-angle dependence and the relevance of the Gamma-couplings.)

Reviewer #3 (Remarks to the Author):

During the past several years, the Kitaev model has been one of the hot topics in the condensed matter physics. This is partially due to the fact that it is exactly solvable with very rich physics including both quantum spin liquid and topological quantum computation. Moreover, it can be potentially realized in real materials with strong spin-orbit coupling, for instance, the “Kitaev materials” including α - RuCl_3 .

In this paper, the authors study the antiferromagnetic Kitaev model on the honeycomb lattice in the presence of magnetic field along the [111] direction. The ground state phase diagram was studied using the variational approach, which is based on the exact fractionalized Majorana-fermion and vison excitations of the pure Kitaev model. In the phase diagram, the authors show that there is a continuous phase transition at a critical magnetic field h_c , which separates the non-Abelian topological phase below h_c and an intermediate phase above h_c . The non-Abelian phase is consistent with previous studies. However, the authors claim that the intermediate phase is a gapped Abelian spin liquid, which is qualitatively distinct with the state reported in previous studies. The results are interesting and the paper is well-written. However, before I can recommend its publication, the authors need to address the following important questions.

The antiferromagnetic Kitaev model has been studied in the past by different groups, which suggest that the intermediate phase above h_c is consistent with a gapless spin liquid, for instance, a $U(1)$ spin liquid with spinon Fermi surface. These previous studies include Zhu et al., PRB 97, 241110 (2018); Gohlke et al., PRB 98, 014418 (2018); Hickey et al., Nature Communications 10, 530 (2019); Patel et al., PANS 116, 12199 (2019). Moreover, the proposed gapless spin liquid was further supported by other independent DMRG calculations including Jiang et al., arXiv:1809.08247, and Jiang et al., PRB 100, 165123 (2019). Both studies directly calculate the entanglement entropy and extract the central charge, i.e., the number of gapless spin mode in the bulk of the system. For instance, Jiang et al., arXiv:1809.08247 reports that there is one gapless mode in the intermediate phase on the 3 leg cylinder. In a separate study, Jiang et al., PRB 100, 165123 (2019) studied an extended Kitaev-Heisenberg model, where they show that the intermediate spin liquid phase of the antiferromagnetic Kitaev model in [111] magnetic field is continuously connected to an enlarged gapless spin liquid region, where a finite number of gapless spin modes was also reported in the bulk of the system.

According to the finite-size scaling in these DMRG studies, it seems likely that the gapless spin mode will survive in the long cylinder limit and the finite-size effect seems negligible. Then an important question comes up immediately, i.e., why the current study did not see the gapless mode, or why these DMRG studies see finite number of gapless spin mode in the bulk of the system? I think that addressing this question is crucial to the current study, as it is the most important discrepancy between the current study and previous studies.

Reply to reviewers comments

Reviewer #1:

“Authors studied the phase transition of antiferromagnetic Kitaev model under the magnetic field perpendicular to the honeycomb plane, [111]-axis. This problem (and extended model) was studied earlier by several others (Refs.37 - 45) using numerical methods or slave particle theories. Authors found an intermediate phase under the field, similar to other previous studies, but the nature of the intermediate phase is different from the earlier claims. They report that the intermediate phase is a gapped spin liquid, which belongs to Kitaev's 16-fold way. The Chern number changes from ± 1 to ± 4 , across the transition around $h_c \sim 0.5$, and it is a continuous transition.

I find the result interesting if true, but unfortunately it is very difficult to read the draft and verify the conclusion. Furthermore, the physical origin of the main conclusion is not clear as the draft is not self-contained. For example, Eq. (3) is a low-energy effective Hamiltonian, but the explanation on how such Hamiltonian is derived in the manuscript is minimal (maybe due to the page limit of the journal). Since authors refer to Ref. 46 (arXiv:2013.13274), which is another paper by the authors, I had to read the reference to get a rough idea on the Hamiltonian. Ref. 46 shows how other interactions such as Heisenberg and Gamma terms lead to various magnetic ordering states.”

We thank the reviewer for the positive comment and for the constructive criticism. Indeed, the variational framework that we are using in this manuscript has been originally introduced in Ref. 46 (now Ref. 48). As the reviewer points out, this new framework was used in the prior work to study the transitions into magnetically ordered states induced by Heisenberg and Gamma terms added to the Kitaev model. In this manuscript we are applying the same variational scheme to answer a more interesting question: what is the new spin liquid state induced by a Zeeman term with the magnetic field along the [111] direction?

We agree with the reviewer that the derivation of the low-energy Hamiltonian was unclear in the original version. Therefore, we have significantly reorganized the paragraph around Eq. (3) [now Eqs. (3) and (4)] to make this derivation more logical and the manuscript more self-contained. In particular, the two terms of the current Eq. (3) directly follow by projecting the pure Kitaev model onto the low-energy subspace: the first term is the matter-fermion problem obtained from the standard exact solution of the model, while the second term accounts for the finite energy of a single flux pair, also discussed in Kitaev's original paper. The remaining terms in Eq. (4) are then obtained by symmetry considerations; each term is the most general symmetry-allowed Hamiltonian that captures the relevant process (the hopping of a flux pair or the hybridization between a flux

pair and a matter fermion). While the parameters q and $p_{R,\alpha}$ in these terms are *a priori* unknown, they are derived in the Supplementary Information by matching the low-energy Hamiltonian with the microscopic one in Eq. (1).

“Few questions need to be addressed, before I make any recommendation.

1. How can one be sure that the gapped spin liquid phase is not preempted by another phase? For example, numerical studies found a putative gapless spin liquid around $h_c \sim 0.44$. It is possible that this gapless phase is different from the author's gapped phase, and preempt the claimed gapped phase. Is there a reason why such a scenario is impossible?”

We thank the reviewer for this question. Although we cannot unequivocally rule out that our proposed phase transition is preempted by a different one (after all, we use a variational approach), we believe that the excellent qualitative and quantitative agreement between our results and previous numerical works (especially Ref. 44) provides very strong evidence for the validity of our conclusions. In particular, we predict a continuous phase transition at approximately the same critical field as the numerical works, and even reproduce some more delicate features of the phase transition, such as the discontinuity in the second derivative of the ground-state energy. Moreover, while all numerical works agree that the intermediate-field phase is a spin liquid, the numerical evidence for the gapless nature of this spin liquid is far from conclusive. Some numerical works speculate that the spectrum should be gapless in the thermodynamic limit, but other works (such as Ref. 44) are more conservative as they are aware of the limitations imposed by size effects (state of the art DMRG/iDMRG methods only allow for cylindrical lattices of circumference not larger than $L_y=5$ unit cells). In particular, the ambiguity between a gapless spectrum and one with a very small gap is revealed by the analysis in Appendix A of Ref. 44.

2. The current theory cannot capture the magnetically polarized phase, the most trivial phase when the field is very large. If so, how one could confirm that the phase does exist? The theory (Eq. 3) may breakdown before h reaches h_c . Could authors quantify a critical strength beyond which the theory does not apply and the gapped phase is below the critical field?

The reviewer is correct that our variational approach is not suitable for studying the second quantum phase transition into the topologically trivial

phase (which is adiabatically connected to the fully polarized state). This is the reason why we are focusing on the first quantum phase transition into the intermediate-field phase. While we cannot establish a rigorous range of validity for our variational approach, we can argue in two different ways that it should break down shortly after the first phase transition, hence correctly capturing the first transition but not the second one.

First, the variational approach is based on the fractionalized excitations of the pure Kitaev model at zero field. These excitations remain well defined throughout the entire low-field phase; however, after the first transition induced by their softening, these original excitations are superseded by the excitations of the new phase, thus invalidating the variational subspace based on the original excitations. We note that this situation is analogous to Ref. 48 where the variational framework was also used to determine the first instability induced by a given perturbation (which was a Heisenberg or Gamma interaction instead of the Zeeman term considered here).

Second, given that our approach is based on flux excitations, defined on top of the flux-free background of the pure Kitaev model (corresponding to $W_p = +1$), we expect that our approach breaks down when the flux expectation value (W_p) becomes much smaller than 1. According to Figure 3c, this happens just after the first transition into the intermediate-field phase.

Finally, we emphasize again that our approach predicts a continuous phase transition at approximately the same critical field as the numerical works and that it reproduces crucial (qualitative and quantitative) features of the numerically observed phase transition.

3. Why the magnetic field acts different from other interactions? Eventually the magnetic field leads to the polarized phase, so if such a possibility (polarized phase) is included together with an incommensurate ordering, can one get an incommensurate (IC) ordering in addition to the gapped spin liquid, or the IC preempts the spin liquid?

This a nice question. To answer it, we must first distinguish between two scenarios: I) strongly first order phase transition; II) weakly first order or continuous quantum phase transition. In the former case, it is not possible to predict the nature of the phase that appears on the other side of the transition. In the second case, the nature of the new phase can be identified

by analyzing the mode that becomes soft for a critical value of the perturbation. In the case of the Heisenberg and Gamma perturbations, the soft mode consists of a *bound state* between a flux pair and a matter Majorana fermion. Since that mode corresponds to a spin excitation, its softening leads to a divergent magnetic susceptibility for some wave vector that signals the onset of magnetic ordering. The situation is qualitatively different for the Zeeman term considered in the present manuscript because the field-induced soft mode is a *single* flux pair *hybridized* with a matter Majorana fermion (i.e., not a bound state). The softening of this mode changes the Chern number of the underlying topologically ordered state. Therefore, the answer to the last question of the reviewer is that one can indeed detect an IC ordering that preempts the spin liquid state by considering different kinds of modes that may soften at a lower value of the critical field. If a *bound state* between a flux pair and a matter Majorana fermion becomes soft at an incommensurate wave vector for a critical field lower than h_c (i.e., the original critical field), we must conclude that the IC ordering preempts the transition into the quantum spin liquid state.

In summary, I found the result interesting, but I cannot recommend the current version of the draft for publication.

We thank the reviewer for the constructive criticism that helped us in improving the presentation of our results.

Reviewer #2 (Remarks to the Author):

The paper aims to provide an interpretation and explanation of previous numerical results concerning the Kitaev model in a magnetic field. Motivated by experimental reports that a magnetic field can suppress magnetic ordering in α -RuCl₃ and that a topological spin liquid may then replace it, several groups had studied the Kitaev model (and its variants) using numerical approaches. These had found an intermediate phase between the Kitaev spin liquid (at zero field) and the magnetically polarized regime.

The present paper presents an analytic (but approximate) treatment of this transition in order to provide insights beyond the numerical findings. Results compare well with numerics and the biggest difference -- gapped excitations, where numerics have not found a gap -- are reasonably

explained by the gaps small size.

The point of this work is that this analytic treatment yields insights that are not easily obtained using numerics (or experiments), e.g. a Chern number of 4 and the fact that it is Abelian. The work appears to be carefully carried out and correct, it is also reasonably clearly described. Given that the study of (extended) Kitaev models continues to be a vibrant field of research, where analytic results are however much rarer than numerical ones, these findings will be of strong interest to readers.

We thank the reviewer for the positive assessment of our manuscript.

It is not so clear to me to what extent the paper will inspire new work beyond the immediate community working on Kitaev models. One thing that could improve the manuscript would be if the conclusions could pick up a theme strongly present in the introduction, namely the experimental research. It would be nice to read a discussion not only of the more immediate relation to the numerical work, but also to experiment. (E.g. the field-angle dependence and the relevance of the Gamma-couplings.)

We agree with the reviewer that an important goal of our manuscript is to motivate experimental efforts toward finding realizations of novel spin liquid states. Following his/her constructive suggestion, we have expanded the conclusions to discuss the stability of the intermediate-field spin liquid and its potential relevance for real materials. According to earlier numerical works (Refs. 38 and 42), the intermediate-field spin liquid is stable against both Heisenberg and Gamma interactions and, in particular, the addition of a ferromagnetic Heisenberg interaction enables a direct field-induced transition between the intermediate-field spin liquid and a zero-field zigzag order (which is the known zero-field ordering of α - RuCl_3 , one of the most promising Kitaev candidate materials). Based on these numerical results, we expect that the intermediate-field spin liquid (which we identify as an Abelian spin liquid with Chern number $C=4$) should actually be easier to find than the low-field spin liquid (i.e., the non-Abelian Kitaev spin liquid with Chern number $C=1$) because it occupies a much larger region of the multi-dimensional phase diagram (see Ref. 42). As we also explain in the new version of the manuscript, the Chern number can be readily extracted from the specific quantized value of the thermal Hall conductivity.

Reviewer #3 (Remarks to the Author):

During the past several years, the Kitaev model has been one of the hot topics in the condensed matter physics. This is partially due to the fact that it is exactly solvable with very rich physics including both quantum spin liquid and topological quantum computation. Moreover, it can be potentially realized in real materials with strong spin-orbit coupling, for instance, the “Kitaev materials” including α - RuCl_3 .

In this paper, the authors study the antiferromagnetic Kitaev model on the honeycomb lattice in the presence of magnetic field along the $[111]$ direction. The ground state phase diagram was studied using the variational approach, which is based on the exact fractionalized Majorana-fermion and vison excitations of the pure Kitaev model. In the phase diagram, the authors show that there is a continuous phase transition at a critical magnetic field h_c , which separates the non-Abelian topological phase below h_c and an intermediate phase above h_c . The non-Abelian phase is consistent with previous studies. However, the authors claim that the intermediate phase is a gapped Abelian spin liquid, which is qualitatively distinct with the state reported in previous studies. The results are interesting and the paper is well-written. However, before I can recommend its publication, the authors need to address the following important questions.

We thank the reviewer for describing the aspects of our work that make it interesting for the broad condensed matter community.

The antiferromagnetic Kitaev model has been studied in the past by different groups, which suggest that the intermediate phase above h_c is consistent with a gapless spin liquid, for instance, a $U(1)$ spin liquid with spinon Fermi surface. These previous studies include Zhu et al., PRB 97, 241110 (2018); Gohlke et al., PRB 98, 014418 (2018); Hickey et al., Nature Communications 10, 530 (2019); Patel et al., PANS 116, 12199 (2019). Moreover, the proposed gapless spin liquid was further supported by other independent DMRG calculations including Jiang et al., arXiv:1809.08247, and Jiang et al., PRB 100, 165123 (2019). Both studies directly calculate the entanglement entropy and extract the central charge, i.e., the number of gapless spin mode in the bulk of the system. For instance, Jiang et al., arXiv:1809.08247 reports that there is one gapless mode in the

intermediate phase on the 3 leg cylinder. In a separate study, Jiang et al., PRB 100, 165123 (2019) studied an extended Kitaev-Heisenberg model, where they show that the intermediate spin liquid phase of the antiferromagnetic Kitaev model in [111] magnetic field is continuously connected to an enlarged gapless spin liquid region, where a finite number of gapless spin modes was also reported in the bulk of the system.

First of all, we thank the reviewer for pointing out two relevant papers that we did not previously cite (the 2018 arXiv by Jiang *et al.* and the 2019 PRB by Jiang *et al.*), especially the second one that shows the remarkable stability of the intermediate-field spin liquid phase.

However, we believe that the numerical evidence for the gapless nature of the intermediate-field spin liquid is far from conclusive. While it is certainly true that some of the above-mentioned works conjecture a gapless spin liquid based on DMRG results on finite lattices, this is not true for all of these works. In particular, the work by Gohlke, Moessner, and Pollmann [PRB 98, 014418 (2018)] does not make any claim relative to this aspect of the problem because the authors acknowledge that their numerical results are not conclusive. This is quite clear from the results that are presented in their Appendix A, as well as from the multiple discussions that we had with two of the authors of that work (M. Gohlke and F. Pollmann). If the system had indeed a gapless spinon Fermi surface, the central charge should take integer values that depend on the number of k_y modes that cross the Fermi surface. However, as it is clear from Appendix A of PRB 98, 014418 (2018), the central charge that results from plotting the entanglement entropy as a function of $\ln\{\xi\}$ does not even take integer values. This is a good indicator that the method is not adequate for determining the nature of the low-energy spectrum (i.e., if it is gapped or gapless). The basic problem of extracting the gap from numerical solutions on finite lattices is that the results are no longer reliable when a characteristic length of the system becomes longer than the smaller linear size of the finite lattice. Since state of the art DMRG/iDMRG methods only allow for cylindrical lattices whose shorter length is no longer than $L_y=5$ unit cells, it becomes clear why the authors of PRB 98, 014418 (2018) have adopted a more conservative attitude. We also note that the different works cited by the reviewer do not agree on the number

of gapless modes that is extracted from the entanglement entropy analysis. Here we quote the work by Patel *et al.*, PANS 116, 12199 (2019):

“However, it seems to be very difficult to obtain a reliable value for c (central charge) unambiguously in a gapless phase using DMRG. Ref. 34 proposes $c=1,0$ using $L_2=3,4$ (number of unit cells along the y -direction) respectively; ref. 33 finds $c =4$ using $L_2=5$; and Ref. 35 calculates $c =1, 2$ using $L_2 =2, 3$ ”

According to the finite-size scaling in these DMRG studies, it seems likely that the gapless spin mode will survive in the long cylinder limit and the finite-size effect seems negligible. Then an important question comes up immediately, i.e., why the current study did not see the gapless mode, or why these DMRG studies see finite number of gapless spin mode in the bulk of the system? I think that addressing this question is crucial to the current study, as it is the most important discrepancy between the current study and previous studies.

Here we remit to our previous answer. In our view, and in the view of some of the above-mentioned numerical experts, the existing numerical results only indicate that the spin gap is either very small or zero. As we explained in the second paragraph of our “Results” section, our theory explains the apparent similarity between the gapped spin liquid that we are finding and the gapless spin liquid with a circular Fermi surface that was proposed in some of the previous numerical studies (see also the low-energy ring of excitations that is depicted in the right inset of Fig. 3a). It is thus important to remark that our results are not in contradiction with existing numerical works. On the contrary, they are in very good agreement with the numerical results presented in PRB 98, 014418 (2018). Therefore, we hope that our work will reopen the discussion on the nature of this remarkable field-induced spin liquid phase.

REVIEWERS' COMMENTS

Reviewer #2 (Remarks to the Author):

I have reread the revised manuscript. The authors have followed my suggestions. I also think that they have sufficiently addressed the other reviewers' concerns by including an extended discussion of the "small gap" vs. "gapless" question.

Reviewer #3 (Remarks to the Author):

My questions have been addressed by the authors, and revisions have been made in the revised manuscript. I am happy with the revised version and thus would like to recommend its publication.